# Online Phase Measurement Profilometry for a Fast-Moving Object

**Jie Gao** **, Yiping Cao \*, Jin Chen and Xiuzhang Huang**

Department of Opto-Electronics, Sichuan University, Chengdu 610064, China;
2018222050046@stu.scu.edu.cn (J.G.); 2018222055221@stu.scu.edu.cn (J.C.); 2018222055222@stu.scu.edu.cn (X.H.)
* Correspondence: ypcao@scu.edu.cn; Tel.: +86-28-85463879

**Abstract:** When the measured object is fast moving online, the captured deformed pattern may appear as motion blur, and some phase information will be lost. Therefore, the frame rate has to be improved by adjusting the image acquisition mode of the camera to adapt to a fast-moving object, but the resolution of the captured deformed pattern will be sacrificed. So a super-resolution image reconstruction method based on maximum a posteriori (MAP) estimation is adopted to obtain high-resolution deformed patterns, and in this way, the reconstructed high-resolution deformed patterns also have a good effect on noise suppression. Finally, all the reconstructed high-resolution equivalent phase shifting deformed patterns are used for online three-dimensional (3D) reconstruction. Experimental results prove the effectiveness of the proposed method. The proposed method has a good application prospect in high-precision and fast online 3D measurement.

**Keywords:** phase measurement profilometry; super-resolution image reconstruction; online three-dimensional measurement; maximum a posteriori; random field

## 1. Introduction

With the development of optics, computer and information technology, 3D measurement technology plays an important role in reverse engineering, industrial 3D testing, medical diagnosis, cultural relic protection and so on [1–3]. The more commonly used 3D measurement technologies are phase measurement profilometry (PMP) [4,5] and Fourier transform profilometry (FTP) [6,7]. Compared with FTP, PMP is a point-to-point phase calculation with multiple frames of deformed patterns, it has higher precision and is more favored by the industry. Common 3D measurement based on PMP requires the fixed position of the object, but the object is moving in online 3D measurement, which will cause the object coordinates in the captured multiple frames of deformed patterns not to correspond, and lead to the error in the PMP phase demodulation. The pixel matching method [8] is used to obtain deformed patterns with consistent object coordinates. Scholars have carried out a lot of research in the field of online 3D measurement. Among them, in order to improve the matching speed, Peng Kuang et al. [9] proposed a new pixel matching method using the modulation of shadow areas in online 3D measurement. To improve the measured precision, Chen Cheng et al. [10] proposed an online phase measuring profilometry for objects moving with straight-line motion. To avoid the loss of phase information due to frequency filtering, Peng Kuang et al. [11] proposed a dual-frequency online PMP method with phase-shifting parallel to the moving direction of the measured object.

Generally, these methods may have a good effect when the speed of the production line is less than 0.2 m/s. However, when the object moves fast online, the captured deformed patterns will appear as motion blur. In order to adapt to the fast speed, the frame rate of the camera can be increased by adjusting the image acquisition mode, but the resolution of captured deformed patterns is bound to be sacrificed simultaneously. In order to obtain high-resolution deformed patterns and improve the measurement precision, we proposed an online phase measurement profilometry method based on super-resolution image

reconstruction [12,13], which combines high-resolution image from multiple frames of low-resolution images with sub-pixel [14] shifts among them. In addition, this paper adopts a super-resolution reconstruction method based on the maximum a posteriori (MAP) [15,16], using Gauss and Markov–Gibbs [17,18] random field models to construct posteriori probability of the high-resolution deformed pattern, the optimal estimation of high-resolution deformed pattern is obtained by minimizing the objective function. In this way, the high-resolution deformed patterns can be obtained. Finally, all the equivalent phase-shifting deformed patterns are demodulated for online 3D reconstruction.

## 2. Principle

### 2.1. Principle of PMP

The sinusoidal grating is designed to parallel to the moving direction, and the projector projects the sinusoidal grating onto the surface of the object, the deformed pattern $I(x,y)$ captured by the camera is:

$$I(x,y) = A(x,y) + B(x,y)\cos(\varphi(x,y) + \delta) \tag{1}$$

where $A(x,y)$ represents the background intensity of the deformed pattern, $B(x,y)$ reflects the contrast of the deformed pattern, $\varphi(x,y)$ is the phase modulated by the height of the object and $\delta$ is the shifting phase. Then in $N$ ($N \geq 3$) step PMP, the $n$-th corresponding deformed pattern is:

$$I_n(x,y) = A(x,y) + B(x,y)\cos\left(\varphi(x,y) + \frac{2n\pi}{N}\right) \tag{2}$$

The phase $\varphi(x,y)$ is calculated by Equation (3):

$$\varphi(x,y) = \arctan\left[\frac{\sum_{n=1}^{N} I_n(x,y)\sin\left(\frac{2n\pi}{N}\right)}{\sum_{n=1}^{N} I_n(x,y)\cos\left(\frac{2n\pi}{N}\right)}\right]. \tag{3}$$

Because $\varphi(x,y)$ is wrapped in $(-\pi, \pi]$ due to the arctan function, it should be unwrapped to be the continuous phase $\Phi(x,y)$ by using the rhombus phase unwrapping algorithm [19], and the object height distribution $h(x,y)$ is reconstructed by the phase-to-height mapping algorithm [20].

$$\frac{1}{h(x,y)} = a(x,y) + b_1(x,y)\frac{1}{\Phi(x,y)} + b_2(x,y)\frac{\Phi_C(x,y)}{\Phi(x,y)} \tag{4}$$

where $a(x,y)$, $b_1(x,y)$, $b_2(x,y)$ can be obtained by plane calibration, $\Phi_C(x,y)$ is the phase of the reference plane, it can be obtained by measuring the reference plane in advance.

### 2.2. Super-Resolution Image Reconstruction Algorithm Based on MAP

2.2.1. The Basic Principles of the MAP

Due to the change of camera acquisition mode for fast online measurement, the captured deformed pattern is degraded to low-resolution deformed pattern, and the main degradation process can be expressed as follows:

$$I''_{n,k} = D_{n,k}B_{n,k}M_{n.k}I''_n + \Delta I''_{n,k} \tag{5}$$

where $I''_{n,k}$ is the low-resolution deformed pattern, which is the $k$-th ($k = 1, 2, \ldots, K$) frame captured in the $n$-th ($n = 1, 2, \ldots, N$) phase-shifting position, and $K$ represents the number of low-resolution deformed patterns captured at the same work station (the same shifting phase), $D_{n,k}$ represents the down-sampling matrix, which means that the high-resolution image is sampled at a certain distance so that the resolution of the image is reduced, $B_{n,k}$ is the blur matrix, it can be used to express the effect of the optical system's blur and aberration on high-resolution image, the point spread function (PSF) is used to express

mathematically. $M_{n,k}$ is the motion matrix, used to characterize the pixel displacement of the low-resolution image relative to the high-resolution image. $I'_n$ is the original high-resolution deformed pattern matrix, and $\Delta I'_{n,k}$ is the additive random noise matrix. $B_{n,k}$ and $D_{n,k}$ of each low-resolution deformed pattern are the same when a reference is selected, but $M_{n,k}$ of each deformed pattern may not be the same because of the motion of the object. For convenience, only one phase-shifting position is analyzed here, so the Equation (5) can be written as:

$$I''_k = H_k I' + \Delta I''_k \tag{6}$$

where $H_k = D_k B_k M_k$ is a degenerate matrix. Only $I''_k$ is known in Equation (6), so we need to use an algorithm to estimate $H_k$ and $\Delta I''_k$ to solve $I'$.

In experiment, to obtain the high-resolution image from the low-resolution image, we need to set the resolution magnification factor $q$ and interpolate the low-resolution image, so that we can interpolate the first frame by bicubic interpolation, and this interpolated image can be regarded as the reference frame of high-resolution image. Then the down-sampling matrix $D_k$ can be solved, and motion matrix $M_k$ between other frames of low-resolution images and the reference frame can be calculated by pixel matching. Blur matrix $B_k$ can be represented by PSF. Above all, $H_k$ is estimated.

Because super-resolution image reconstruction is an ill-posed problem, $I'$ needs to be estimated, so what satisfies the condition is a set of images, not the unique solution. However, under the Bayesian theory [21], MAP can flexibly add the prior probability of image, which is the mathematical expression of image features, and then it can convert ill-posed problems into well-posed problems by means of regularization. Finally, the unique solution of the problem can be obtained.

MAP estimation refers to the maximum probability of obtaining a high-resolution image under the condition of known low-resolution image sequence $I''$, which can be expressed as solving $\max P(I'|I'')$. According to Bayesian theory, it can be calculated as:

$$\hat{I} = \mathrm{argmax} P\left(I' \middle| I''\right) = \mathrm{argmax}\left(\frac{P(I''|I')P(I')}{P(I'')}\right) \tag{7}$$

where $\hat{I}$ represents the estimation of a high-resolution image $I'$, $P(I'|I'')$ is a posteriori probability, $P(I''|I')$ is a conditional probability when a high-resolution image degenerates into a low-resolution image, $P(I')$ and $P(I'')$ are the prior probabilities of the high-resolution image and the low-resolution image, respectively. Since $P(I'')$ has no effect on the solution of $\hat{I}$, it can be omitted and the Equation (7) is reduced to:

$$\hat{I} = \mathrm{argmax}(P(I''|I')P(I')) \tag{8}$$

Logarithmically available

$$\hat{I} = \mathrm{argmax}(\log P(I''|I') + \log P(I')) \tag{9}$$

Or

$$\hat{I} = \mathrm{argmin}(-\log P(II''|I') - \log P(I')) \tag{10}$$

Equation (10) is the initial form of the objective function. In order to solve it, the prior probability $P(I')$ and the conditional probability $P(I''|I')$ must be determined. And their distributions depend on assumptions about the statistical model of the image

According to the degeneration model $I''_k = H_k I' + \Delta I''_k$, a low-resolution image can be regarded as a random field and the mean of the random field is $H_k I'$ because of the random noise, so Gauss random field can be used to solve $P(\hat{I}|I')$. Markov random field describes the local statistical properties of images, while Gibbs random field describes the global properties by joint probability, so they are combined by equivalence, and the global statistical results can be calculated by using the local Gibbs distribution model.

2.2.2. Establishment of Objective Equation

When the image statistical model is selected, Equation (10) can be written as

$$\hat{I} = \text{argmin}[\sum_{k=1}^{K} \| I''_k - H_k \hat{I} \|^2 + \alpha \sum_{c \in C} \rho_\alpha(di_c)] \tag{11}$$

where $\sum_{k=1}^{K} \| I''_k - H_k \hat{I} \|^2$ represents the difference between the actual data and the estimated value, $\sum_{c \in C} \rho_\alpha(di_c)$ is the regular term, and $\alpha$ is the coefficient of the regular term, which determines the influence of the regular term on the image estimation. $C$ is all the cliques [22] in the image matrix neighborhood system, $\rho_\alpha(x)$ is the potential function related to the cliques $C$, different potential functions can be selected to obtain different texture statistical properties, and the parameter $di_c$ is the variance of its pixel mean. So the objective function can be expressed as:

$$\Theta(\hat{I}) = \sum_{k=1}^{K} \| I''_k - H_k \hat{I} \|^2 + \alpha \sum_{c \in C} \rho_\alpha(di_c) \tag{12}$$

The function of the regular term is to constrain the estimation $\hat{I}$ according to the desired goal, and reduce the deviation from the optimal solution. The regular term used in this experiment is a function of the Gibbs distribution model, which describes the energy of a feature in the image neighborhood. The higher the energy, the lower the probability of feature will appear. According to the objective equation, the higher the energy of the feature, the larger the regular term, the greater the inhibition in the process of minimizing the objective equation. Therefore, in order to reduce the influence of noise on the estimation $\hat{I}$, the feature should reflect the difference between the noise and the original image and make the noise larger. The potential function $\rho_\alpha(di_c)$ is chosen according to the penalty degree of removing features of image, such as the Huber equation and linear equation.

2.2.3. Iterative Solution

Since it is hard to directly solve $\hat{I}$ when $\Theta(\hat{I})$ is minimum, we can choose to estimate $\hat{I}$ iteratively by using the gradient descent [23], and projecting the negative gradient of the objective function into the constraint space at each iteration, thus $\hat{I}$ converges to the local minimum and to the global minimum as far as possible.

The whole iterative process can be described as:

1. firstly, the low-resolution image is interpolated by bicubic interpolation, and the initial estimation $\hat{I}^0$ is obtained. When $m = 0$, where $m$ is the number of iterations, the mean square error (*MSE*) of the initial negative gradient $MSE_0$ is calculated.
2. calculate the gradient of the objective function $g_m = \nabla\Theta(\hat{I}_m)$.
3. calculate the projection map to the constraint space $p_m$, $p_m = -Pg_m$, where $P$ is the projection operator, and $P = E - H_k^T (H_k H_k^T)^{-1} H_k$, where $E$ is the identity matrix.
4. using the gradient of the objective function to constrain $\hat{I}_m$, and the expression is

$$\hat{I}^{m+1} = \hat{I}^m + (-\beta p_m) \tag{13}$$

   where $\beta$ is the learning rate, which is the coefficient of the negative gradient of the objective function.
5. calculate the MSE of the negative gradient of this iteration $MSE_m$. If $\frac{MSE_m}{MSE_0} \le \varepsilon$, $\hat{I}^m$ is the best estimation, where $\varepsilon$ is the iteration termination flag. Otherwise, return to step $2°$.

   *MSE* is defined as:

$$MSE = \frac{1}{L_1 L_2} \sum_{x,y} \left[\hat{I}(x,y) - I'_r(x,y)\right]^2 \tag{14}$$

where $L_1$, $L_2$ are the width and height of the high-resolution deformed pattern. $I'_r$ is the reference of high-resolution deformed pattern, which is obtained by interpolating the first frame of low-resolution images. The whole process of MAP is shown in Figure 1.

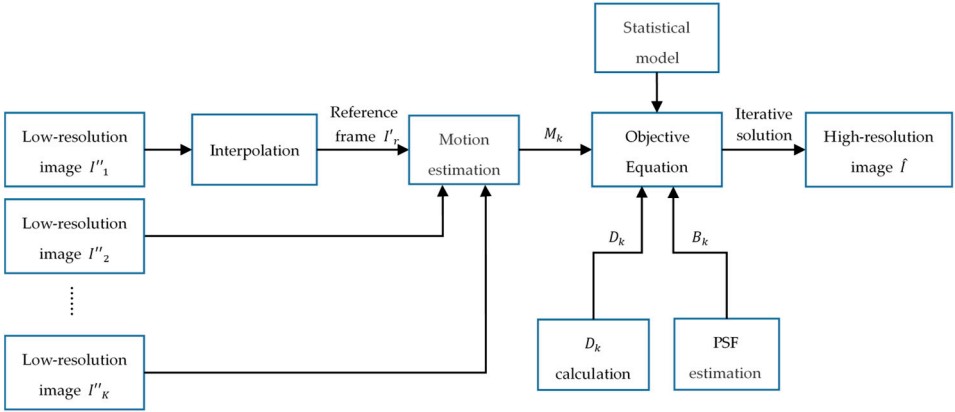

**Figure 1.** The flow-process diagram of maximum a posteriori (MAP).

### 2.3. Equivalent Deformed Patterns

Using the previous MAP super-resolution image reconstruction method, we can reconstruct a set of high-resolution deformed patterns in different positions of an online moving object. In order to obtain equivalent deformed patterns, which means that an object in each deformed pattern is in the same position, this paper adopts Fast 3D measurement based on improved optical flow for dynamic objects by Peng Kuang et al. [24] for pixel matching. As shown in Figure 2, Figure 2a is the modulation of the deformed pattern $I_1'$, Figure 2b is the modulation of the deformed pattern $I_N'$. Figure 2a is used as a reference, the motion displacement of the object is calculated by pixel matching of Figure 2a,b; at the same time, according to the calculated motion displacement, the modulation with the same position of the object is obtained, as shown in Figure 2d, where the black area is the part of the missing information after the left shift of the pixel matching; phase demodulation can be carried out for the region of interest (ROI), which is the dotted line region in Figure 2c,d. $I_N'$ is moved in the reverse direction according to the calculated motion displacement, and the equivalent deformed patterns $I_1$ and $I_N$ can be obtained by intercepting $I_1'$ and $I_N'$ on the ROI in Figure 2d. Similarly, by matching the modulation of $I_2' - I_{N-1}'$ with the modulation of $I_1'$, and intercepting on the ROI in Figure 2d, a set of equivalent deformed patterns $I_1, I_2, I_3, \ldots, I_N$ can be obtained. The phase $\varphi(x, y)$ of the online object can be obtained by substituting $I_1, I_2, I_3, \ldots, I_N$ into Equation (3). The continuous phase $\Phi(x, y)$ of the online object can be obtained by using the rhombic phase unwrapping algorithm and 3D shape of the online object can be obtained by substituting $\Phi(x, y)$ into Equation (4).

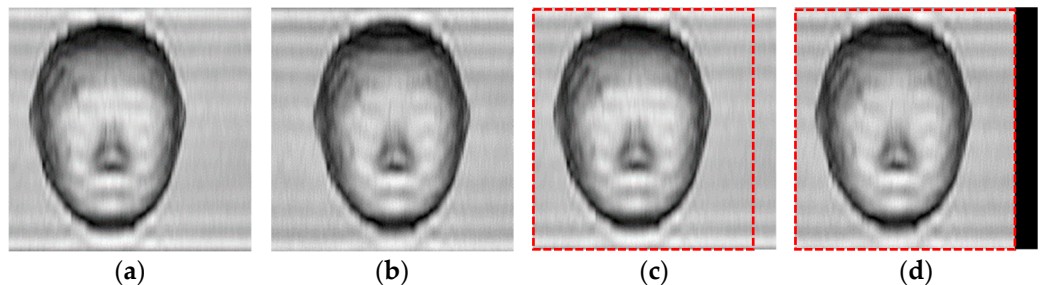

|     |     |     |     |
| --- | --- | --- | --- |
| (**a**) | (**b**) | (**c**) | (**d**) |

**Figure 2.** Modulation of high-resolution fringe pattern: (**a**) Modulation of $I_1'$; (**b**) Modulation of $I_N'$; (**c**) Modulation of $I_1'$ with ROI; (**d**) Modulation of $I_N'$ with ROI. ROI: region of interest.

## 3. Experiment and Analysis

In order to verify the feasibility and practicability of the proposed method, an online 3D measurement experimental system is built. As shown in Figure 3, the projector used in this experiment is PLED-W200 DLP, and the image collector is a SDI-C2010M camera, the highest frame rate of this camera is 60 fps when the image pixel size is 1920 × 1080; in order to adapt the fast speed, we increase the frame rate to 100 fps while the pixel size is decreased to 256 × 256. During this measurement, the object is driven from left to right by the electric translation platform Y100SC01 at a fast speed, Through the previous experimental calibration, we measured that the relationship between pixels and the motion distance is approximately 1.245 mm/pixel. The entire image acquisition process is:

1.  The designed 5 frames of sinusoidal gratings with a shifting phase of $2/5\pi$ are combined into a repeated video.
2.  The projector projects the video onto the object.
3.  Start the electric translation platform at a given speed.
4.  Turn on the DLP frame synchronization signal to trigger the camera, thus achieving synchronous acquisition.
5.  After 0.2 s, turn off the DLP frame synchronization.

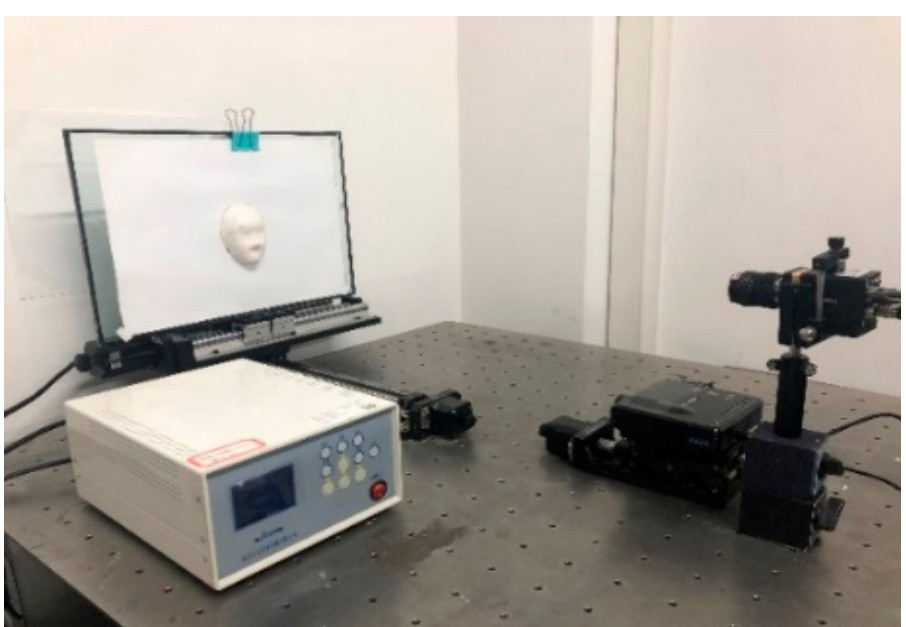

**Figure 3.** Experimental system.

In this experiment, the frame rate of the video is 25 fps, and the frame rate of the camera is 100 fps. Thus, at the same work station, 4 frames ($K = 4$) of deformed patterns are captured. The image acquisition is real time, data processing is post-processing, and the data processing includes super-resolution reconstruction, obtaining equivalent deformed patterns and PMP 3D reconstruction. Because the iterative solution of the super-resolution reconstruction takes some time, and iteration time is related to hardware and data process mode, in future, we will use a better computer or adapt graphics processing unit (GPU) acceleration to speed up data processing.

The proposed method is compared with the averaging method, and the experimental data are shown in Figure 4. Figure 4a is a measured object, which is a "face" model, and in this experiment, the speed of the object is 0.2 m/s. Figure 4b is a set of low-resolution deformed patterns captured at the first work station, which are $I''_{11}$, $I''_{12}$, $I''_{13}$, $I''_{14}$, and the pixel size is 256 × 256.

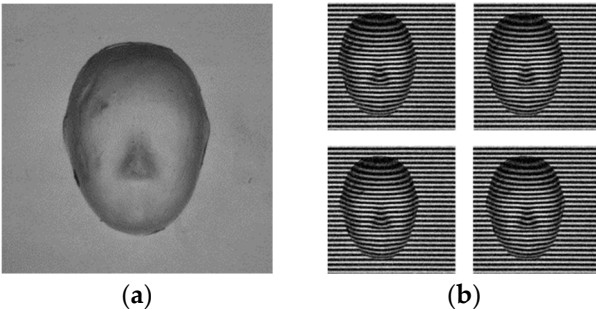

**Figure 4.** Measured data of "face" model: (**a**) Measured object; (**b**) A set of deformed patterns.

The reconstructed deformed patterns are shown in Figure 5, Figure 5a is $I''_{11}$, and it is regarded as a reference frame. The process of pixel matching is in Figure 5a,b. The points marked in Figure 5a are the selected feature points of $I''_{11}$, the other points marked in Figure 5b are the matching points of $I''_{14}$. The difference between the two set of points represents the movement of the "face". Then the estimated motion matrix, blur matrix and down-sampling matrix are used for subsequent iterations. In this experiment, the super-resolution factor $q$ is 2, the max number of iterations $m$ is 200. Figure 5c,d are the reconstructed results of the averaging method and MAP, respectively, and their pixel size is 512 × 512. By comparison, it can be seen that the deformed pattern reconstructed by MAP is clearer, and there is no large motion blur. At the same time, compared with the original low-resolution deformed pattern, the deformed pattern reconstructed by MAP has a good effect on noise suppression. Figure 5e,f are the enlargements of the rectangular areas of Figure 5c,d, respectively. Compared with Figure 5e, Figure 5f has a clearer edge, which further indicates that noise is reduced effectively using MAP.

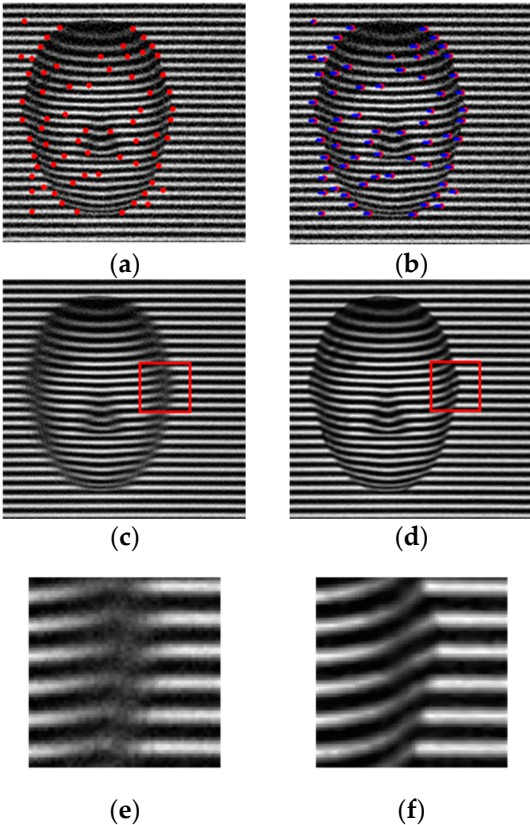

**Figure 5.** Low-resolution deformed patterns and super-resolution reconstructions: (**a**) Reference; (**b**) $I''_{14}$ with matching points; (**c**) Reconstruction by averaging; (**d**) Reconstruction by MAP; (**e**) Enlargement of rectangular area in (c); (**f**) Enlargement of rectangular area in (d).

In the same way, high-resolution deformed patterns of the other four work stations can also be obtained, and the reconstructed results are shown in Figure 6. Figure 6a shows five frames of high-resolution deformed patterns reconstructed by the averaging method, and Figure 6c shows five high-resolution deformed patterns reconstructed by the proposed method, which are $I'_1, I'_2, I'_3, I'_4, I'_5$. In Figure 6a,c, the marked dotted line shows that the position of the object in each frame is not the same. We adopt optical flow [24] to obtain equivalent deformed patterns, and the results are shown in Figure 6b,d. The pixel size of Figure 6b,d is 458 × 512, and in Figure 6b,d, the position of the object in each frame is the same. Figures 6e and 6f are, respectively, the experimental results of PMP 3D reconstruction using Figure 6b,d. From Figure 6e,f, it can be seen that the reconstructed result by the proposed method is better than that by the averaging method, and the phase information is more complete. At the same time, in order to better analyze the measuring results of the proposed method, the measuring result of eight-step PMP [25] is taken as the quasi truth value, and the eight-step PMP is taken for the same but stationary object.

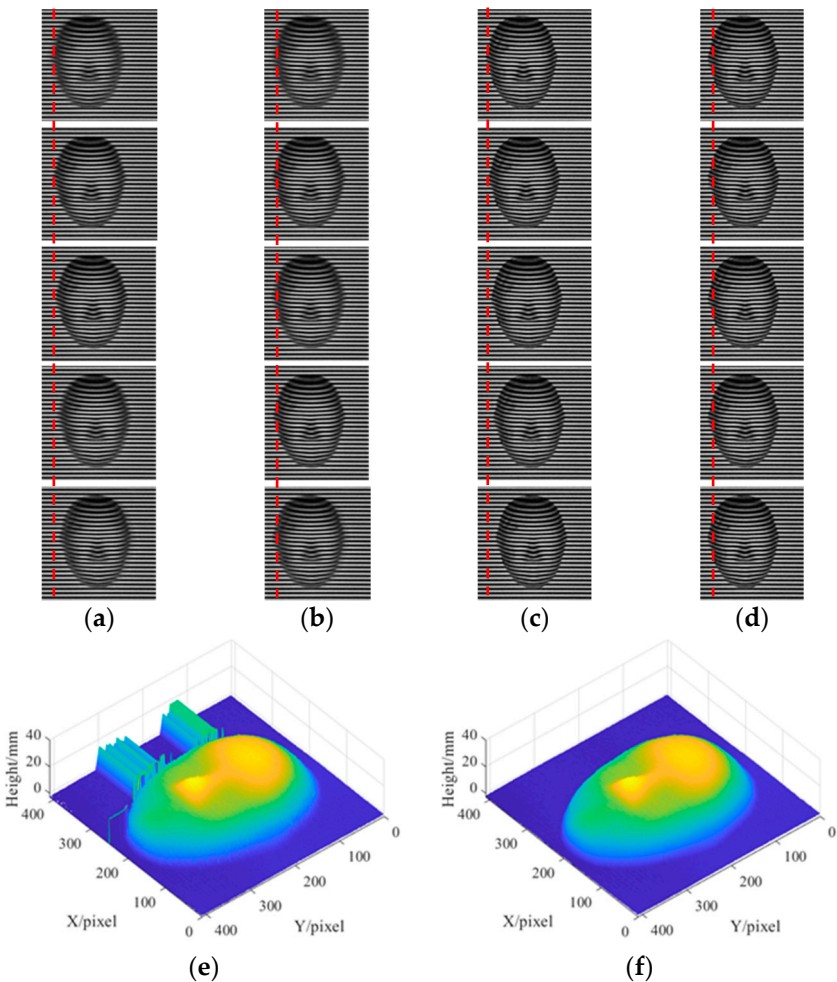

**Figure 6.** Experimental results for "face" model, pixel size of Figure 6a,c is 512 × 512, pixel size of (**b,d**) is 458 × 512: (**a**) High-resolution deformed patterns by averaging; (**b**) Equivalent deformed patterns by averaging; (**c**) High-resolution deformed patterns by the proposed method; (**d**) Equivalent deformed patterns by the proposed method; (**e**) Object reconstruction by averaging; (**f**) Object reconstruction by the proposed method.

Figure 7 shows the 255th column sectional view of the reconstructed 3D profile of the object with different methods.

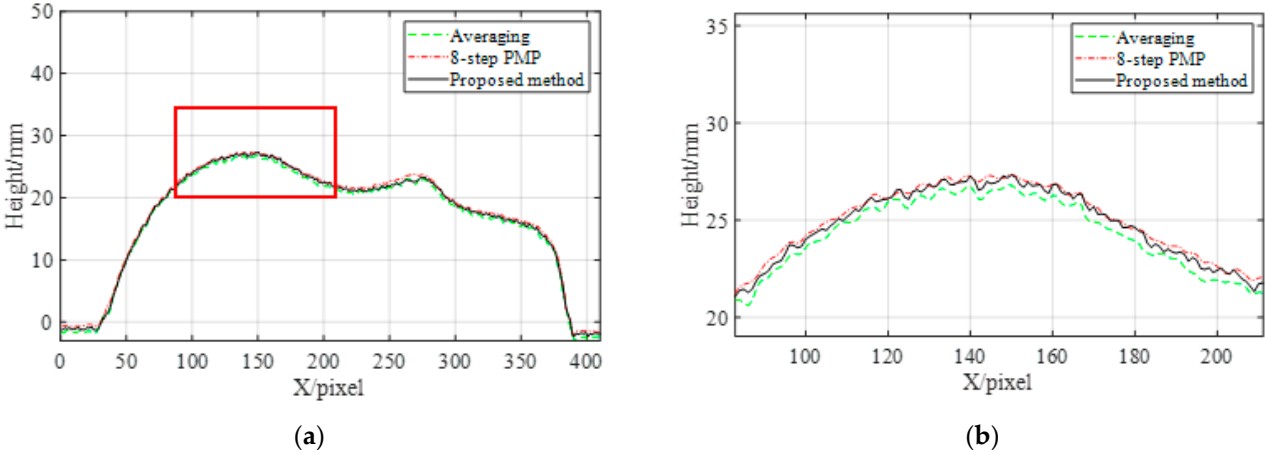

(**a**)                (**b**)

**Figure 7.** The 255th column data of the reconstructed object: (**a**) Sectional view of the 3D profile of the object; (**b**) Magnified view of the forehead area.

Figure 7b is the enlargement of Figure 7a forehead (rectangular area). The dot-and-dash line is the result of the eight-step PMP, the dotted line is the result of the averaging method, and the solid line is the result of the proposed method. As can be seen from Figure 7a, the reconstructed result of the proposed method is closer to that of the eight-step PMP, and better than that of the averaging method. As can be seen from the details shown in Figure 7b, the results of the proposed method are also very close to those of the eight-step PMP. The reconstructed result of the proposed method is better than that of the averaging method, either in the whole or in the detail. The experimental results indicate that the proposed method can improve the resolution while preserving the details of the object.

In order to further verify the applicability of the proposed method, this experiment used a more complex model, a "snail" model, and the speed of the object is 0.5 m/s, super-resolution $q = 2$, the max num of the iterators $m$ is 200. The experimental data and the results of the comparison experiment are shown in Figure 8. Figure 8a, Figure 8b, and Figure 8c are one frame of the high-resolution deformed patterns obtained by the averaging method, proposed method and eight-step PMP, respectively. Figure 8d–f show the corresponding wrapped phases, and they are obtained by calculating all equivalent deformed patterns, and through phase unwrapping and height mapping, the reconstructed results are shown in Figure 8g–i. It also can be seen that motion blur in Figure 8b is less than that in Figure 8a; The wrapped phase in Figure 8e is clearer than that in Figure 8d and is closer to that in Figure 8f. By comparing Figure 8g–I, the height reconstruction by the proposed method is obviously better than that by the averaging method, and is closer to that by eight-step PMP.

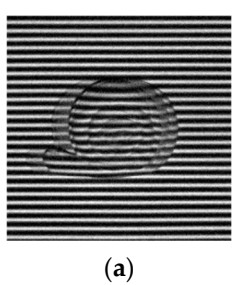 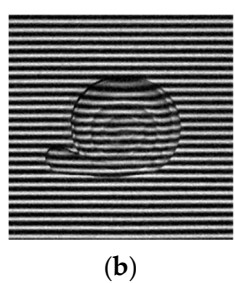 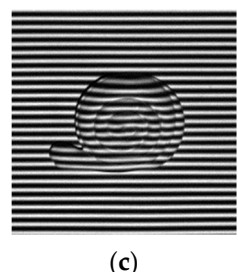

(**a**)              (**b**)              (**c**)

**Figure 8.** *Cont.*

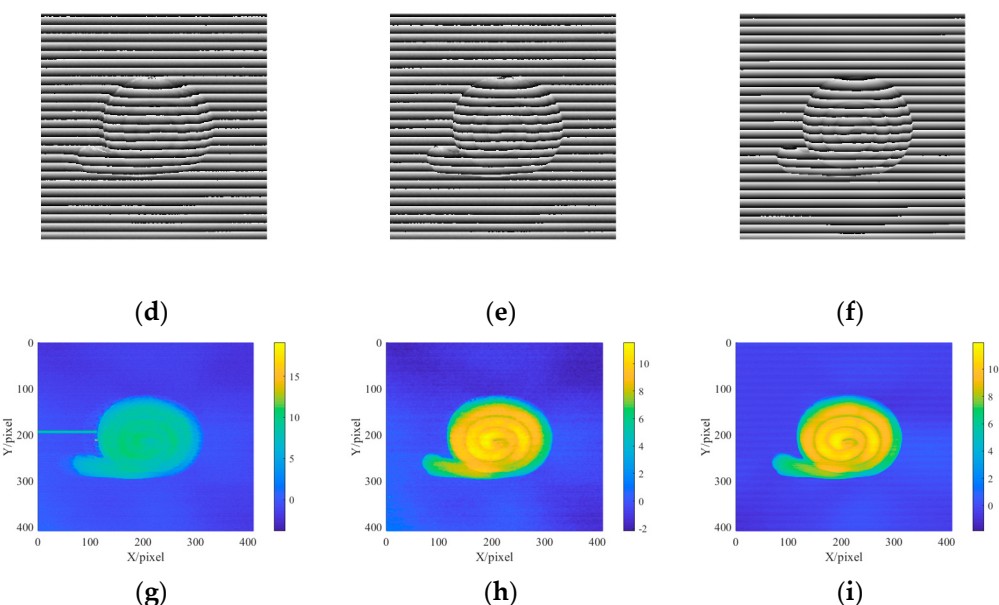

**Figure 8.** Experimental data and the comparison experiment results of the ''snail'' model: (**a**) High-resolution deformed pattern by averaging; (**b**) High-resolution deformed pattern by the proposed method; (**c**) High-resolution deformed pattern by 8-step PMP; (**d**) Wrapped phase by averaging; (**e**) Wrapped phase by the proposed method; (**f**) Wrapped phase by 8-step PMP; (**g**) Reconstruction by averaging; (**h**) Reconstruction by the proposed method; (**i**) Reconstruction by 8-step PMP.

To further analyze the details, Figure 9 is a cross-sectional comparison of the three methods; the dot-and-dash line is the result of the eight-step PMP, the dotted line is the result of the averaging method, and the solid line is the result of the proposed method. It also can be seen that the reconstructed result of the proposed method is better than that of the averaging method, either in the whole or in the detail. It means that the proposed method has a higher precision than the averaging method. The experimental results as shown in both Figures 8 and 9 prove that the proposed method has a good effect on the more complex object.

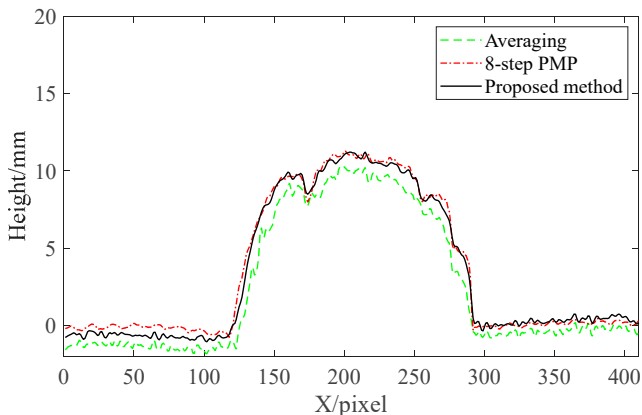

**Figure 9.** The 255th column data of the reconstructed object.

To further quantitative analysis of the error of the proposed method and the averaging method, we measured the heights of the different planes. The heights of the planes are known and measured by metrological grating, and they are taken as the truth values. The heights are 5 mm, 10 mm and 15 mm. According to the proposed method and the averaging method, we take an online 3D measurement of known height plane. Then we use root mean squared error (RMSE) to analyze the errors. The results of measurements are shown in Table 1. RMSE1 and RMSE2 are the calculated RMSEs of the averaging

method and the proposed method, respectively. For example, in the experiment of 5 mm, the measured RMSE of the averaging method is 0.358 mm, and that of the proposed method is 0.091 mm. The results prove that the proposed method has a higher precision and higher reliability than the average method.

**Table 1.** Error analysis of planes with different heights (/mm).

| Height | RMSE1 | RMSE2 |
| --- | --- | --- |
| 5.000 | 0.358 | 0.091 |
| 10.000 | 0.317 | 0.074 |
| 15.000 | 0.336 | 0.086 |

### 4. Conclusions

In the online 3D measurement of a fast-moving object, the frame rate of a camera can be adjusted to adapt to the deformed pattern capturing of the fast-moving object, and the resolution of camera will be sacrificed. In this paper, an online phase measurement profilometry method for a fast-moving object was proposed, and the experimental results prove that the proposed method can not only improve the resolution of the deformed patterns, but also ensure to restore most of the details of the object. The proposed method can adapt to the online 3D measurement when the fast-moving object moves at the speed of 0.5 m/s. We also think that the proposed method has a good application prospect in high-precision and fast online 3D measurement. However, the proposed method is mainly limited by the frame rate of the camera, and if a higher frame rate camera is selected, the speed can be higher. In later work, we may adopt a super-resolution image reconstruction method based on deep learning, the training model can monitor and train more factors.

**Author Contributions:** Conceptualization, Y.C. and J.G.; methodology, Y.C.; validation, J.G., J.C. and X.H.; investigation, J.C., X.H. and J.G.; resources, Y.C.; writing—original draft preparation, J.G.; writing—review and editing, Y.C. and J.G.; visualization, J.G.; project administration, Y.C. All authors have read and agreed to the published version of the manuscript.

**Funding:** This work is supported by the Special Grand National Project of China (under grant No. 2009ZX02204-008).

**Institutional Review Board Statement:** Not applicable.

**Informed Consent Statement:** Not applicable.

**Data Availability Statement:** Data is contained within the article. The data presented in this study are available in Online Phase Measurement Profilometry for a Fast-Moving Object.

**Conflicts of Interest:** The authors declare no conflict of interest.

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
