# Peer review of "Online Phase Measurement Profilometry for a Fast-Moving Object"

_applsci, doi:10.3390/app11062805_

Round 1

Reviewer 1 Report

The article discusses how improve the image detection on an object in fast movement. The analysis of the problem and the proposed solution are illustrated in detail. The state of the art is well exposed, with sufficient and pertinent references cited.

Principle and methodology are suitably presented, while in the discussion of application and results the figures could be improved.

Especially because the discussion concerns an image analysis and image elaboration, the images should clearly illustrate the steps of the image processing evidencing the differences among the compared methods and among the various steps of the elaboration.

If it is possible, please improve the resolution of the presented figures, or enlarge them or evidence inside the picture the details that change during the elaboration or that differ between two images.

In particular, in the captions of the multiple images (e.g. Fig. 5, Fig. 7) it would be useful to add explanations about the differences that are almost invisible among (a), (b), (c), (d).

The conclusions report almost the same concepts expressed in the abstract. The meaning, usefulness, scope could be further discussed in the conclusions. As regards the application of the proposed methodology, it could be interesting to discuss possible extensions or limitations. Future improvements are not mentioned in the conclusions and should be.

English is correct and the text is clear.

Reviewer 2 Report

The work as presented in unsuitable for publication for the following reasons:

1) The way in which the material is exposed is extremely poor.  Many sentences do not make any sense and/or are very long and hard to understand.  See, for example, the paragraph at lines 102-106 and the first sentence of the Abstract.  The choice of words is often incomprehensible, to the point that sentences do not convey any information.  Many terms are not defined.  What does it mean to "purify a deformed pattern"?  What is it meant by "fuzzy matrix B_{n, k}"?

2) It is not clear what problem the authors are trying to solve.  One reason for this is provided in the previous point.  Another reason is because the authors do not provide enough background information.  A clearly-described diagram would also help.

3) The performances and benefits of the propose method are not assessed objectively.  In fact, lines 250-270 contain many occurrences of words such as "closer", "better", "very close", "clearer", which convey no useful information.

Reviewer 3 Report

The team demonstrated online PMP for a fast-moving object. To image a fast-moving object, a spatial resolution of the image should be compromised given the limited image-acquisition bandwidth of the camera. The reduced spatial resolution inevitably degrades the quality and accuracy of PMP. The team estimates high-resolution images from measured low-resolution images using a MAP method – the ill-post problem was solved under the known priori probability of the image based on Bayes’ Theorem. The proposed method seemingly works, but there are few points to be clarified.

- Although the degenerate matrix, H_k, includes image operations (down-sampling, motion blur, aberrations, etc), this priori information is not included in the estimation process. It is rather considered as a totally unknown operator. In this manner, seemingly this method simply does up-sampling of the image, noise-reduction operation, and pixel matching. What is the fundamental difference? The author might add more detailed explanations.

- How many iterations are required for the estimation to converge? How is the computational burden?

- In this experiment, the moving direction of the object is parallel to the fringe. How is the image when it is orthogonal? Does it provide a similar performance?

- In Figures 6, 8, authors might provide a quantitative value to evaluate the performance. i.e. structural similarity index, or mean-squared-error assume 8-step PMP provides a ground-truth value.

- How is the performance of this method compared to the pixel matching method [ref 5]?

- Missing color bars in Fig.7. Typo, line 267

Reviewer 4 Report

I have carefully read the paper “Online Phase Measurement Profilometry For Fast-Moving Object.” The authors present an application of the super-resolution technique for phase measurement profilometry (PMP). The increased resolution provides a possibility to increase the frame rate of the registration and therefore it becomes possible to work with fast-moving objects.  

The paper is well structured but lacks in the delivery of the results. Here are my major comments:

  1. Entire sections 2.2.1 and 2.2.2 need references otherwise it provides a wrong feeling that the authors have developed all the described techniques, but I believe it is not.
  2. It is not clear why matrix in the Eq. (5) depend on the number of frame k and n. Should not be PSF (as authors say for Bn,k), down-sampling D, and motion matrix the same for all K and N?
  3. Line 90 – what is the meaning for “work station”?
  4. What are the clusters and potential function in line 136?
  5. Section 2.2.3. Iterative solution needs clarification in:
  6. Step 1. Which interpolation algorithm is used for initial estimation?
  7. Step 3. In the calculation of P, what is the E?
  8. Step 3. In the calculation of P, before in line 101, the authors wrote Hk should be estimated, but here it is written as already known and they provide no estimation for it.
  9. Step 5. The stated stop condition is appropriate for the simulation when I’ is known, but what authors use for the experimental data?
  10. Authors should provide more information about the used setup, for example, I could not find what is the ‘electric translation platform Y10SC01’. Or I have found the camera ‘SDI-C2010M’ [1] but the highest frame rate there is 60 fps contrary to the 100 fps stated in the paper.
  11. The description of the experiment in lines 204-208 is not clear. 
  12. Is true reconstruction by the 8 steps PMP taken for a moving or stationary object?
  13. Authors do not provide any information about measurement and processing time. From the given information, it is impossible to make a conclusion about possible online reconstruction. I can estimate only that for a given 100 fps and 0.5 m/s speed the movement distance during exposure is 5 mm, but it is not clear how it relates to pixels of the camera and therefore how complex the problem is.
  14. Paper is about super-resolution but authors provide neither a super-resolution factor nor pixel sizes.
  15. To have a complete investigation, I encourage authors to make additional experiments to find the limits of their technique by estimation of reconstruction error depending on different speeds of an object.

 Additionally, the paper needs clarification in minor comments:

  1. Lines 26-27 authors wrote: “…patterns, which has higher precision and is more favored by the industry” but it is not clear to which technique they are comparing.
  2. Why K is taken equal to 4??
  3. Figures with colors need color bars. Axis should have titles and units.
  4. Stacks of images (as e.g. Fig.3(b) or Fig.5(a-d)) do not bring useful information.
  5. Reference 15 is not full.
  6. The paper has a lot of long sentences (e.g. lines 217-224) which are very hard to follow. Split them, please. 

Considering all statements described above, I recommend the paper for a major revision.

References:

[1] https://www.imperx.com/wp-content/uploads/2017/10/Cheetah_C2010_3G-HD-SDI_user-manual_rev1.0.pdf

Round 2

Reviewer 2 Report

I have read the manuscript multiple times in the hope that the latest version would convey useful information.  While I noticed an improvement, many deficiencies make me believe that this work is still not suitable for publication.

My main concerns are listed below.

1) My suggestion of including a diagram was completely ignored by the authors.  Besides valuable in and of itself, the authors can use the diagram to clarify their notation by appropriate labeling parts of the diagrams with the variables and functions they use in their equations.

2) Language and style is still a major problem.  The way in which the material is exposed puts too much effort on the reader, as they are often forced to wonder about the meaning of specific words.  As an example, the meaning of the word "purify" is still not explained in the paper.  As far as I can tell, that word is not standard in the literature.  The authors should not expect to satisfy this reviewer just by explaining to him what they mean by "purify", thus without correcting the problem in the paper.  The authors are writing for their target audience, not for the reviewer.

3) Many sentences are still unclear or confusing.  As an example, the first paragraph in the introduction is about 3/4 of a page long, and it covers the motivation of the paper, literature review as well as a brief description of the proposed method.  Often times, there no natural transition from a sentence to the next, and the authors keep jumping from topic to topic.

I strongly urge the authors to completely rewrite the paper and restart the submission process.

Reviewer 4 Report

The authors have made appropriate corrections in the manuscript, therefore, I think, it might be published. 

Round 3

Reviewer 2 Report

Although extensive editing of English language and style are required before publication, the paper now provides all the technical information that was missing before.